# TEMPORAL STRIDE AS AN INDUCTIVE BIAS FOR SEMANTIC LEARNING IN MEDICAL VIDEOS

## MEANINGFULNESS STATEMENT

We consider a meaningful representation of life to be one that focuses on stable, underlying structure instead of visual fluctuations. In medical videos, this means capturing anatomical context and pathology that remain consistent over time, despite changes in camera motion, lighting, or viewpoint. Our work supports this idea by showing that a simple choice in temporal sampling can strongly influence what a self-supervised encoder learns. By using sparse temporal sampling, the model in this setting ignores high-frequency visual noise and instead learns representations that reflect clinically meaningful structure without relying on manual annotations.

## ABSTRACT

Self-supervised video representation learning has become a standard paradigm in medical imaging, yet the semantic implications of the temporal sampling stride remain unexplored. In this work, we investigate whether the temporal spacing between training frames acts as an inductive bias that shapes the learned representations. We conduct a controlled study using VideoMAE v2 on full-length colonoscopy videos, comparing encoders pre-trained with dense sampling (stride 1) versus sparse sampling (stride 30). Our analysis reveals a fundamental trade-off: dense sampling yields representations dominated by low-level appearance and motion features, resulting in a polyp detection F1-score of 60.3% and a near-random anomaly detection AUC of 0.449 when using a Euclidean distance. Conversely, sparse sampling forces the encoder to capture anatomical semantics, improving F1-score by 16.5 percentage points and achieving near-perfect anomaly detection under Mahalanobis scoring (AUC = 0.998).

## 1 INTRODUCTION

Self-supervised learning methods such as VideoMAE (Tong et al., 2022) are now widely used for pre-training video encoders without manual annotations. This is particularly appealing in medical imaging, where expert labeling is costly and time-consuming (Hirsch et al., 2023).

Despite growing adoption of these methods, one basic design choice has received relatively little attention: *how does the temporal spacing between sampled frames influence what the model learns?* In most prior work, temporal stride is treated as an empirical hyperparameter, often inherited from action recognition benchmarks (Carreira and Zisserman, 2018). The implicit assumption is that denser sampling provides richer supervision. For medical video, however, this assumption may not hold.

In endoscopic videos, rapid frame-to-frame variations arise from camera motion, lighting changes, specular reflections, and motion blur but at the same time, clinically relevant semantic content such as anatomical location or the presence of pathology tends to remain stable over much longer temporal windows (Wang et al., 2024). The choice of temporal stride therefore determines which of these signals dominates the training data presented to the model.

This distinction becomes clear when considering masked autoencoding under different sampling regimes. With dense sampling, where stride 1 is used and consecutive frames are approximately 33 ms apart, neighboring frames are highly redundant and change only minimally. In such case, masked patches can often be reconstructed using local texture and illumination cues from adjacent frames (Fu et al., 2024). In contrast, with sparse sampling, where stride 30 corresponds to frames

one second apart, consecutive frames may depict different anatomical regions as the endoscope advances, reducing the effectiveness of local copying strategies and placing greater emphasis on persistent anatomical structure and semantic context.

Based on this observation, we hypothesize that temporal stride acts as an implicit inductive bias: dense sampling favors representations dominated by appearance-level features, whereas sparse sampling encourages learning of higher-level semantic features that remain stable (Barbed et al., 2025) and visible, with only slight changes in shape or location, across larger temporal gaps.

To test this hypothesis, we conduct a controlled study on colonoscopy videos from the REAL-Colon dataset (Biffi et al., 2024). Based on the success of general-purpose encoders on endoscopic tasks (Taha and Lukmanov, 2025b;a), we train VideoMAE v2 encoders that differ only in temporal stride and evaluate the resulting representations using downstream tasks and embedding analyses.

Our results show a consistent trade-off between appearance sensitivity and semantic discrimination as a function of temporal stride, suggesting that temporal sampling plays a central role in shaping the semantic content of self-supervised video representations. For domains in which meaningful signals evolve slowly relative to visual nuisance variation, sparse temporal sampling provides a simple and effective inductive bias during pre-training.

## 2 METHOD

### 2.1 PROBLEM FORMULATION

Let $\mathcal{V} = \{I_1, I_2, \ldots, I_T\}$ denote a video sequence of $T$ frames captured at frame rate $f$ (30 FPS in our setting). For self-supervised pre-training, we construct training clips $\mathbf{x} \in \mathbb{R}^{C \times \tau \times H \times W}$ by sampling $\tau = 16$ frames with temporal stride $s$:

$$\mathbf{x} = \{I_t, I_{t+s}, I_{t+2s}, \ldots, I_{t+(\tau-1)s}\}. \tag{1}$$

Each clip therefore spans a temporal receptive field of

$$T_{\text{clip}} = \frac{(\tau - 1)s}{f} \text{ seconds.} \tag{2}$$

For stride $s = 1$, this corresponds to $T_{\text{clip}} \approx 0.5\text{s}$, whereas for $s = 30$, $T_{\text{clip}} \approx 16\text{s}$. This large difference in temporal coverage is central to our analysis.

### 2.2 EVALUATION TASKS

We evaluate clip embeddings using linear probes for polyp detection and brightness classification, and we compute distances from healthy embeddings using Euclidean and Mahalanobis metrics (details in Appendix A.3).

### 2.3 EMBEDDING SPACE STRUCTURE

To visually confirm these findings, Figure 2 presents UMAP projections of the test embeddings.

For the Dense model (left), pathological clips (red) are inextricably entangled with healthy ones (blue). This confirms that the dense representation relies on low-level appearance cues (such as lighting or texture) that are shared between healthy and pathological tissue. For the Sparse model (right), pathological clips self-organize into distinct, compact clusters or like islands. This clustering emerges purely from the self-supervised objective without any class labels.

## 3 RESULTS

### 3.1 LINEAR PROBE PERFORMANCE

Table 1 summarizes linear-probe classification performance for both encoders. The Dense encoder achieves 60.3% F1 on polyp detection, indicating limited separability of pathological versus healthy

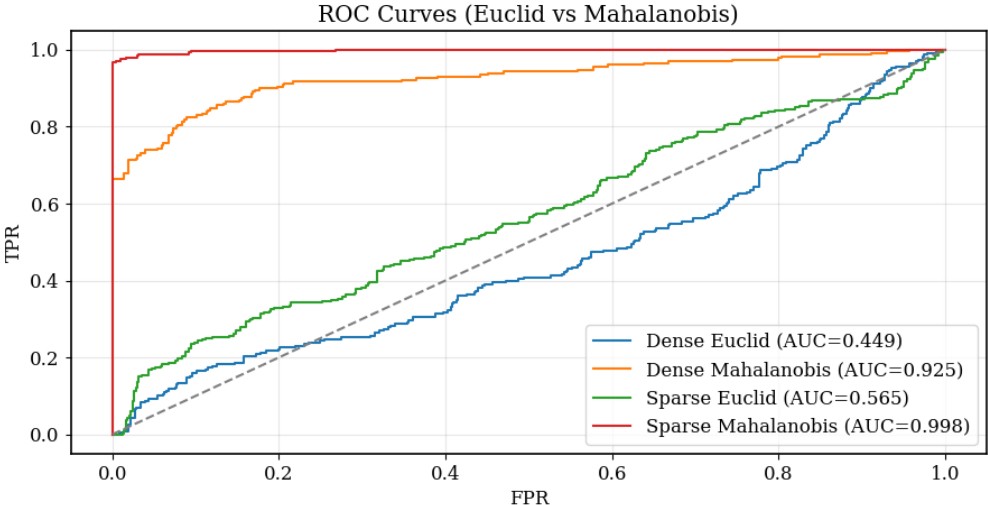

Figure 1: **Anomaly Detection Performance.** ROC curves comparing the ability of Mahalanobis distance to separate polyp clips from healthy ones. Using Euclidean distance, the Dense encoder (blue) yields an AUC of 0.449, while Mahalanobis distance yields AUC = 0.925, while the Sparse encoder achieves AUC = 0.998 under Mahalanobis scoring.

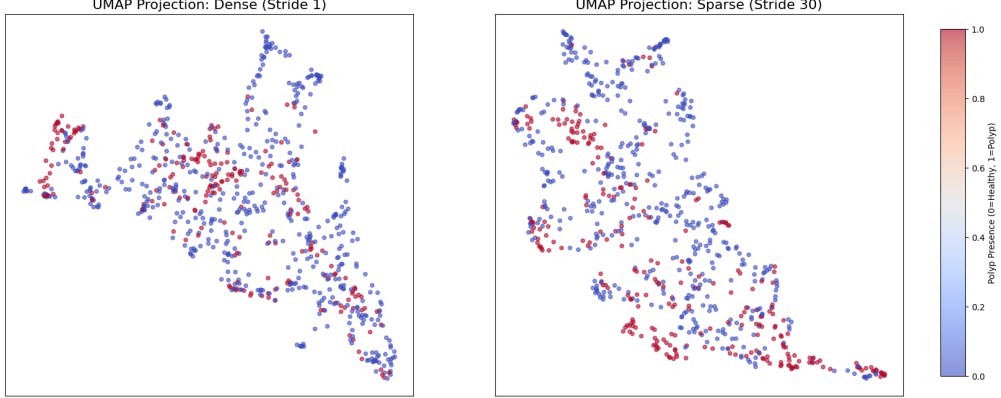

Figure 2: **Latent Space Organization.** UMAP projections of test videos embeddings colored by ground-truth pathology (Red=Polyp, Blue=Healthy). **Left (Dense):** Pathological clips are scattered and mixed with healthy ones. **Right (Sparse):** Pathological clips form distinct compact clusters.

clips. The Sparse encoder reaches 76.8% F1 which is a 16.5 percentage point improvement. This pattern is reversed for brightness classification: Dense performs better on this appearance task (96.8% vs 95.5%), while Sparse shows lower accuracy (invariance) to lighting conditions.

Table 1: Linear probe classification performance. Dense sampling yields slightly higher accuracy on appearance tasks, while sparse sampling significantly improves semantic pathology detection.

| Encoder | Polyp Detection (F1) | Brightness (Binary Acc.) |
|---|---|---|
| Dense (stride 1) | 0.603 | 0.968 |
| Sparse (stride 30) | 0.768 | 0.955 |
| Δ (Sparse - Dense) | +0.165 | -0.013 |

## 3.2 ANOMALY DETECTION

Figure 1 shows anomaly scores for both encoders on test videos containing a polyp segment. Using Euclidean distance, the Dense encoder produces a noisy signal with frequent spikes during healthy navigation, responding to events such as camera motion and lighting changes, and achieves a ROC-AUC of 0.449. Using the same Euclidean score, the Sparse encoder improves to AUC = 0.565. In contrast, when using Mahalanobis distance, the Dense encoder reaches an AUC of 0.925, while the Sparse encoder achieves near-perfect separation AUC = 0.998.

For the sparse encoder, distances remain low and stable during healthy segments, then rise sharply and persistently when the polyp appears. The ROC-AUC of 0.998 demonstrates almost perfect discrimination, suggesting that sparse sampling leads to embeddings where pathology dominates the latent organization rather than being obscured by appearance variation.

## 3.3 EMBEDDING SPACE STRUCTURE

Figure 2 visualizes 2D UMAP projections of embeddings, colored by pathology labels (left) and brightness bins (right). For Dense, polyp clips are scattered and heavily overlap with healthy ones, while brightness forms distinct clusters. This confirms that Dense representations are primarily appearance-driven.

For Sparse, polyp clips form compact clusters separated from healthy tissue, indicating semantic organization. Brightness gradients remain, showing some retention of appearance information, but clinical content dominates the structure.

## 3.4 DISCUSSION

These results support our hypothesis that temporal stride acts as an inductive bias controlling the semantic level of learned representations. Dense sampling ($s = 1$) allows the model to solve the reconstruction task via local appearance high-frequency visual features. Sparse sampling ($s = 30$) breaks temporal redundancy, which forces the model to rely on semantic structure as the difference between frame $i$ and frame $i + 30$ is high.

For medical video tasks such as pathology detection (classification) or anatomical localization (dense prediction tasks), sparse sampling offers clear advantages despite the intuition that denser frame coverage is better. The 16.5% F1 improvement on polyp detection and near-perfect anomaly discrimination highlight the impact of this simple design choice.

### 3.4.1 LIMITATIONS

We consider only two stride values, intermediate rates need to be tested. Our experiments are restricted to colonoscopy videos, and effects may differ in other domains. Finally, our analysis relies on linear probes and distance metrics and using one encoder, which capture only certain aspects of representation quality. Future work should evaluate and work on other medical imaging modalities.

## 4 CONCLUSION

This work highlights the role of temporal stride in self-supervised medical video learning. We demonstrate that stride is not only a hyperparameter for computational efficiency. Our experiments show that dense sampling biases the model towards high-frequency visual nuisance variables such as lighting and camera motion, while sparse sampling attends more to semantic meaningful representations of anatomical structure. These results suggest that future efforts in medical AI must carefully align temporal sampling strategies with the clinical target where sparsity prioritize for pathology detection while dense sampling could be reserved for high-frequency tasks such as instrument tracking.

ACKNOWLEDGMENTS OF AI USE

We used an AI language model to assist with polishing parts of the text. The authors bear full responsibility for the content.

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

## A    APPENDIX

### A.1    APPEARANCE VS SEMANTIC FEATURES

We distinguish between two types of visual information commonly present in endoscopic video:

- **Appearance features** $\mathbf{a}_t$: high-frequency variations such as lighting changes, blur, and texture that vary rapidly between frames

- **Semantic features** $c_t$: low-frequency content such as anatomical structure and pathology that remains stable over longer temporal windows

We hypothesize that when the temporal stride $s$ is small, the inter-frame signal is dominated by changes in appearance, $\Delta \mathbf{a} = \mathbf{a}_{t+s} - \mathbf{a}_t$. As $s$ increases, appearance varies too strongly to provide reliable reconstruction cues, and successful reconstruction increasingly depends on semantic features $c_t$ that persist across time.

## A.2 TRAINING PROTOCOL

We train VideoMAE v2 (Wang et al., 2023) models using the standard masked autoencoding objective. For each input clip $\mathbf{x}$, we randomly mask a subset $\mathcal{M}$ of spatiotemporal tubes (90% masking ratio) and train an encoder $f_\theta$ and decoder $g_\phi$ to minimize the reconstruction loss

$$\mathcal{L}_{\text{MAE}} = \frac{1}{|\mathcal{M}|} \sum_{p \in \mathcal{M}} \left\| \mathbf{x}_p - g_\phi\big(f_\theta(\mathbf{x}_{\setminus \mathcal{M}})\big) \right\|_2^2, \tag{3}$$

where $\mathbf{x}_{\setminus \mathcal{M}}$ denotes the input with masked patches removed.

We train two models with identical architectures (ViT-Small, 12 layers, embedding dimension $d = 768$) and identical optimization settings, differing only in temporal stride:

- **Dense sampling**: $s = 1$ (consecutive frames)
- **Sparse sampling**: $s = 30$ (frames sampled one second apart)

Training is performed on approximately $400\,\text{K}$ frames sampled from the REAL-Colon dataset over 20 epochs using the AdamW optimizer (Loshchilov and Hutter, 2019), batch size 16, and a learning rate of $1.5 \times 10^{-4}$ with cosine decay (Loshchilov and Hutter, 2017).

## A.3 EVALUATION TASKS

After pre-training, we freeze the encoders and extract clip-level embeddings $\mathbf{z} = f_\theta(\mathbf{x}) \in \mathbb{R}^d$, obtained by mean-pooling all patch tokens. Evaluation is performed on held-out test videos. We train logistic regression classifiers on frozen embeddings for two downstream tasks:

- **Polyp detection** (semantic): binary classification where a clip is labeled positive if any of its constituent frames contain a polyp bounding box.
- **Brightness classification** (appearance): Binary classification based on binned mean frame intensity.

We model the distribution of healthy tissue embeddings as a multivariate Gaussian $\mathcal{N}(\boldsymbol{\mu}_h, \boldsymbol{\Sigma}_h)$ estimated from clips without polyps. For each test embedding $\mathbf{z}$, we compute the Mahalanobis distance (Lee et al., 2018)

$$D_M(\mathbf{z}) = \sqrt{(\mathbf{z} - \boldsymbol{\mu}_h)^\top \boldsymbol{\Sigma}_h^{-1} (\mathbf{z} - \boldsymbol{\mu}_h)}. \tag{4}$$

We evaluate the ability of $D_M$ to discriminate polyp clips from healthy clips using ROC-AUC. For completeness, we also report Euclidean distance from the healthy centroid as a simple baseline. The underlying assumption is that semantically meaningful representations will place pathological clips far from the healthy distribution, whereas appearance-dominated representations will not.

## A.4 DATASET AND IMPLEMENTATION DETAILS

We use REAL-Colon (Biffi et al., 2024), a dataset of full-length colonoscopy procedures with frame-level polyp bounding boxes provided in Pascal VOC XML format. All frames are resized to $224 \times 224$ pixels.

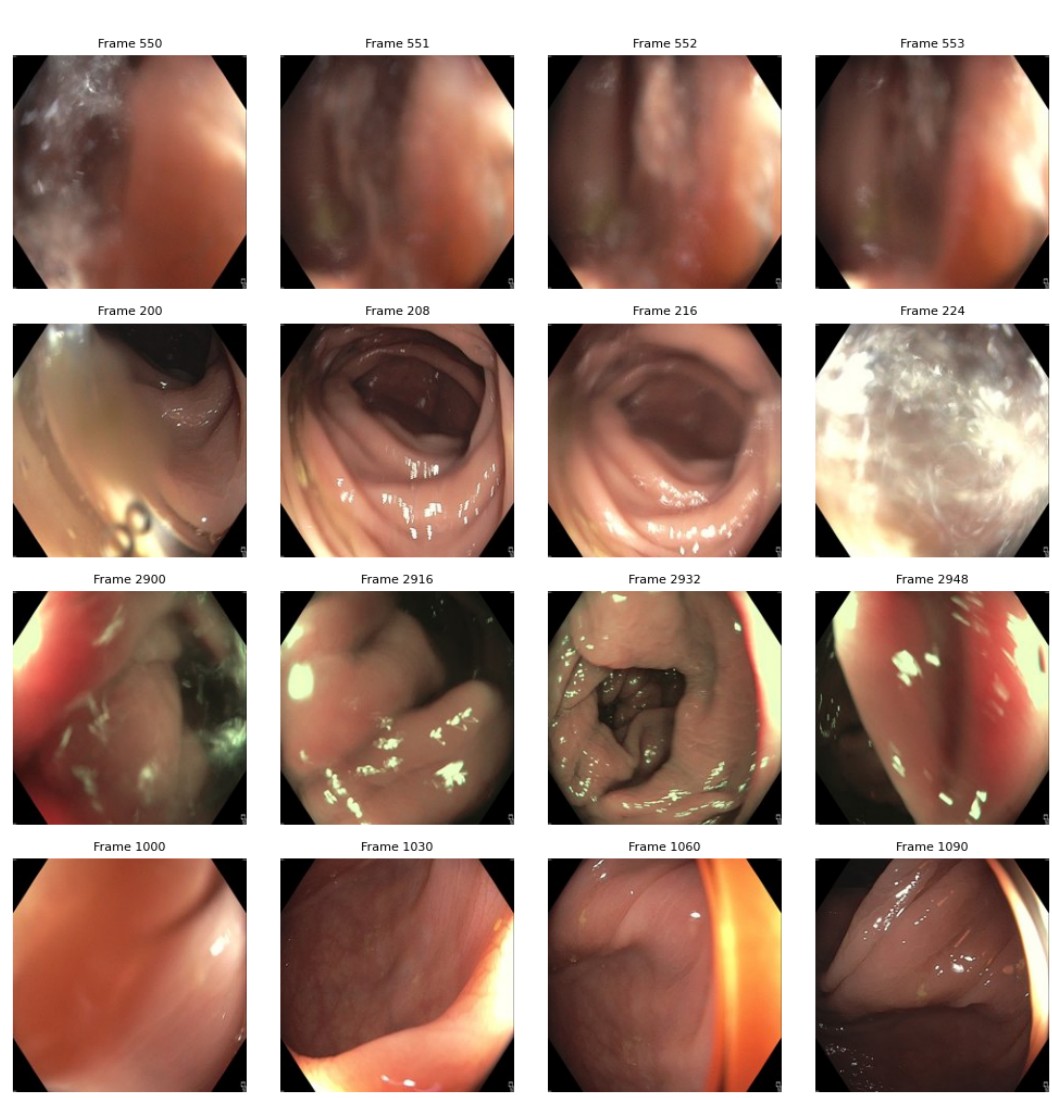

Figure 3: REAL-colon frames sampled at different time steps.

