# OpenReview forum: "TEMPORAL STRIDE AS AN INDUCTIVE BIAS FOR SE- MANTIC LEARNING IN MEDICAL VIDEOS"
_ICLR.cc/2026/Workshop/LMRL — Submitted to ICLR 2026 Workshop LMRL_

### Official Review · Reviewer_RFuh · 2026-02-23
**essentially an investigation of an existing architecture VideoMAE2 to medical colonoscopy videos**

**Rating:** 5
**Confidence:** 4

**Review:**

The authors investigate whether the temporal spacing between training frames acts as an inductive bias that shapes the learned representations. They conduct a controlled study using VideoMAE v2 on full-length colonoscopy videos, comparing encoders pre-trained with dense sampling versus sparse sampling. Their analysis reveals a key tradeoff: dense sampling yields representations dominated by low-level appearance and motion features, sparse sampling forces the encoder to capture anatomical semantics andf improves the F1 scores. This is essentially an investigation of an existing architecture VideoMAE2 to medical colonoscopy videos.

---

### Official Review · Reviewer_Bo8T · 2026-02-24
**The insight that dense sampling can hurt semantic learning by exploiting texture and lighting cues is a useful observation. Yet, this effect is already well-established in VideoMAE literature. While its extension to medical video analysis is valuable, it remains underexplored in the paper.**

**Rating:** 4
**Confidence:** 5

**Review:**

This paper studies the effect of temporal stride in self-supervised pre-training for medical video analysis, focusing on polyp detection and brightness classification. The authors compare dense sampling vs sparse sampling and analyze the resulting embeddings using linear probes, Euclidean/Mahalanobis distances, and UMAP visualizations.

- The choice of temporal stride as an inductive bias is clearly justified, linking temporal coverage to semantic versus appearance-level feature extraction. Sparse sampling improves polyp detection and achieves near-perfect binary anomaly detection, demonstrating practical significance for medical video tasks.

- The downstream tasks (polyp detection and brightness) are relatively straightforward. It would be more robust to see if sparse sampling also improves complex tasks like depth estimation, tool tracking, anatomical localization, or multi-class pathology detection where temporal continuity (dense sampling) is usually considered vital.

- It’s unclear if the sparse/dense sampling conclusions generalize across other medical video datasets, anatomical structures, or longer/shorter videos. As noted in the limitations, comparing only s=1 and s=30 is quite extreme. The work lacks an ablation study on the optimal stride. The study is restricted to colonoscopy videos. In other medical videos (e.g., echocardiograms where the heart beats rapidly), a stride of 30 might skip over the motion needed to identify pathology, potentially making the findings less applicable to high-speed physiological motion.

- In Figure 2, the difference in UMAP projections is subtle rather than significant. Neither projection shows clean separation, in both cases, red and blue are largely intermixed, which suggests the embeddings don't strongly encode polyp presence as a primary axis of variation.

Although the authors note several limitations, including two stride values, a focus on colonoscopy, and reliance on a single encoder with linear probes, the work requires more rigorous analysis.

---

### Meta-Review · Area_Chair_7mAn · 2026-02-28

**Recommendation:** Reject
**Confidence:** 3

**Metareview:**

This is a nice direction to explore & I think the authors should continue to work on it, but the conclusions rest on comparing only two stride lengths. If you could show a continuous change as function of the stride length (e.g. comparing stride lengths `[1, 7, 15, 23, 30]`) you could build a much more convincing case.

---

### Decision · Program_Chairs · 2026-03-02

**Decision:**

Reject

**Comment:**

Please see the meta-review.